# Validation and Optimization of a Stable Isotope-Labeled Substrate Assay for Measuring AGAT Activity

**DOI:** 10.3390/ijms252312490

**Published:** 2024-11-21

**Authors:** Alex Lee, Lucas Anderson, Ilona Tkachyova, Michael B. Tropak, Dahai Wang, Andreas Schulze

**Affiliations:** 1Department of Biochemistry, University of Toronto, Toronto, ON M5S 1A1, Canada; alexa.lee@sickkids.ca; 2Genetics and Genome Biology, The Hospital for Sick Children, Toronto, ON M5G 1X8, Canadailona.tkachyova@sickkids.ca (I.T.); mbt@sickkids.ca (M.B.T.); dahai.wang@sickkids.ca (D.W.); 3Department of Pediatrics, University of Toronto, Toronto, ON M5S 1A1, Canada

**Keywords:** enzyme assay, AGAT activity, creatine deficiency syndromes, isotope-labeled

## Abstract

L-arginine: glycine amidinotransferase (AGAT) gained academic interest as the rate-limiting enzyme in creatine biosynthesis and its role in the regulation of creatine homeostasis. Of clinical relevance is the diagnosis of patients with AGAT deficiency but also the potential role of AGAT as therapeutic target for the treatment of another creatine deficiency syndrome, guanidinoacetate N-methyltransferase (GAMT) deficiency. Applying a stable isotope-labeled substrate method, we utilized ARG 15N_2_ (ARG-δ2) and GLY 13C_2_15N (GLY-δ3) to determine the rate of 1,2-13C_2_,15N_3_ guanidinoacetate (GAA-δ5) formation to assess AGAT activity in various mouse tissue samples and human-derived cells. Following modification and optimization of the assay, we analyzed AGAT activity in several mouse organs. The K_m_ and V_max_ of AGAT in mouse kidney for GLY-δ3 were 2.06 mM and 6.48 ± 0.26 pmol/min/mg kidney, and those for ARG-δ2, they were 2.67 mM and 2.17 ± 0.49 pmol/min/mg kidney, respectively. Our results showed that mouse kidneys had the highest levels of enzymatic activity, followed by brain and liver, with 4.6, 1.8, and 0.4 pmol/min/mg tissue, respectively. Both the heart and muscle had no detectable levels of AGAT activity. We noted that due to interference with arginase in the liver, performing the enzyme assay in liver homogenates required the addition of Nor-NOHA, an arginase inhibitor. In immortalized human cell lines, we found the highest levels of AGAT activity in RH30 cells, followed by HepaRG, HAP1, and HeLa cells. AGAT activity was readily detectable in lymphoblasts and leukocytes from healthy controls. In our assay, AGAT activity was not detectable in HEK293 cells, in human fibroblasts, and in the lymphoblasts of a patient with AGAT deficiency. Our results demonstrate that this enzyme assay is capable of accurately quantifying AGAT activity from both tissues and cells for diagnostic purposes and research.

## 1. Introduction

Creatine homeostasis is essential for maintenance of cellular energy metabolism. The creatine-/phosphocreatine-/creatine kinase system connects the intracellular sites of ATP production and utilization [1,2] and functions as a temporal and spatial energy buffer and regulator of cellular energetics [3,4]. The cellular concentration of creatine is tissue-specific and stably maintained by an interplay of cellular creatine uptake and intracellular creatine biosynthesis.

L-arginine: glycine amidinotransferase (AGAT; EC 2.1.4.1), encoded by the *GATM* gene, is the rate-limiting enzyme in creatine biosynthesis. In humans, AGAT is mainly expressed in the liver, kidney, pancreas, gastrointestinal tract, and brain [5,6,7]. Piglets reveal high AGAT enzyme activity in kidney and pancreas [8]. In the rat pancreas, the precursor of creatine, guanidinoacetate (GAA) synthesized by AGAT, accounts for approximately 15% of daily creatine loss [9], while renal GAA synthesis reflecting AGAT activity is almost equivalent to the daily renal creatine loss [10].

The enzymatic synthesis of GAA from L-arginine and L-glycine was first reported by Borsook et al. [11]. AGAT is a promiscuous enzyme, and many compounds can serve as acceptors of the amidino group from L-arginine such as 4-aminobutyric acid (forming 4-guanidinobutyric acid), lysine (forming homoarginine), 5-aminovaleric acid (forming 5-guanidinovaleric acid), 3-aminopropionic acid (forming 3-guanidinopropionic acid), ethanolamine (forming guanidinoethanol), and taurine (forming 2-gunidinoethanesulfonic acid). Glycine has the lowest K_m_ (5.0 mM) and highest V_max_ (39.8 µmol/mg protein/h) [12], therefore favoring GAA synthesis and by extension creatine synthesis.

With the discovery of creatine deficiency syndromes (CDS) [13], the enzymatic activity of AGAT gained clinical and academic interest due to its central role in creatine biosynthesis and its regulation. While AGAT is one of the causes of CDS (AGAT deficiency), it is also a therapeutic target for another CDS, guanidinoacetate N-methyltransferase (GAMT) deficiency [14]. In the latter condition, the toxic accumulation of GAA can be reduced by pharmacological reduction of either AGAT enzyme activity or AGAT gene expression.

CDS are caused by biallelic pathogenic variants in either the *GATM* or *GAMT* genes or through mutations in *SLC6A8* on the X-chromosome which encodes for the creatine transporter [13]. Patients with CDS have severely reduced levels of creatine within the body, particularly in the brain, and suffer from developmental delays, intellectual disabilities, behavioral abnormalities, speech problems, and seizures [15].

There are several methods in which suspected patients with CDS are diagnosed. One of these methods is with in vivo magnetic resonance spectroscopy which is used to determine the abundance of creatine in localized regions within the brain [16,17,18]. Another method may involve the use of liquid chromatography-tandem mass spectroscopy (LC-MS/MS) to measure the levels of GAA, creatine, and creatinine in urine, blood, and cerebrospinal fluid [17]. These measurements may provide information to help identify the underlying cause of the condition. For example, patients with low levels of GAA and creatine may have deficiencies in AGAT while individuals with high levels of GAA but low levels of creatine might have GAMT deficiency [17]. Currently, most patients are identified or suspected through abnormal findings in CDS genes by genome-wide sequencing. However, if these diagnostic techniques provide ambiguous or uncertain results, additional supplementary methods or assays may be necessary to confirm or exclude a diagnosis. One such assay is an enzymatic assay that uses isotope-labeled compounds to measure the activity of AGAT [19,20,21]. These labeled compounds can act as a substrate in a reaction and the labeled product that is produced can be quantified via LC-MS/MS. The rate of product generated correlates with enzyme activity.

AGAT enzyme activity has previously been measured from various sources such as human and rat kidney homogenates as well as in several cell lines [19,20,21]. While there are several methods that can be used to quantify AGAT activity, many of them can be complicated due to their use of radioisotopes [22] or may result in overestimations of enzymatic activity. In the latter scenario, these enzyme assays use ornithine as a readout for AGAT activity [23]. However, since ornithine can be produced by other enzymes, such as arginase 1 (ARG1), this may result in AGAT enzyme activity appearing to be higher than expected. As such, in order to accurately measure AGAT activity, we adapted a stable isotope-labeled substrate assay from Verhoeven et al. [21]. In this assay, L-[guanido- ^15^N_2_] arginine (ARG-δ2) and [U-^13^C_2_,^15^N] glycine (GLY-δ3) are used to generate [1,2-^13^C_2_,^15^N_3_] GAA (GAA-δ5) (Figure 1). This method differs from previously published methods in that AGAT activity is determined by measuring the concentration of GAA-δ5 produced, which can only be produced if both labeled substrates (ARG-δ2 and GLY-δ3) are present. Furthermore, since there is no other enzyme known that produces GAA from arginine and glycine, this assay provides greater accuracy in quantifying AGAT activity.

In this study, our goal was to validate and optimize the original protocol from Verhoeven et al. to determine and establish baseline levels of AGAT activity from various sample types, including mouse organs, immortalized cell lines, and patient-derived cells.

## 2. Results

### 2.1. GAA-δ5 Is Produced Exclusively in the Presence of Both GLY-δ3 and ARG-δ2

To validate that the stable isotope-labeled substrate assay worked, we performed a reaction in which unlabeled substrates (GLY + ARG) or labeled substrates (GLY-δ3 + ARG-δ2) were added to mouse kidney homogenates and measured the peak intensity for GAA and GAA-δ5 on extracted ion chromatograms (Figure 2). We determined that under baseline conditions with no substrates added, GAA was still detectable, which suggested that the tissue inherently contained GAA. When unlabeled substrates are added, GAA levels increase significantly which indicates that additional GAA is being produced. With the addition of labeled substrates, GAA levels are still detected, but the peak intensity is similar to that of the baseline. When we looked at the levels of GAA-δ5 being produced, we observed that the peak only occurred in the presence of both labeled substrates and was not present under the baseline or unlabeled substrates conditions. Based on these observations, it can be concluded that the assay is able to produce detectable levels of GAA-δ5 in the presence of labeled substrates.

Taking this assay a step further, we carried out the reaction in mouse kidney homogenate with four different combinations of substrates: GLY + ARG, GLY-δ3 + ARG, GLY + ARG-δ2, and GLY-δ3 + ARG-δ2. In addition, we also performed the assay with both unlabeled and both labeled substrates in the presence of ornithine which is a known inhibitor of AGAT [24] (Figure 3). For all reactions, we measured the levels of GAA, GAA-δ5, and ornithine produced. We observed that the formation of GAA only occurred with both unlabeled substrates while GAA-δ5 was only produced when both labeled substrates were present (Figure 3A). In situations where one unlabeled and one labeled substrate was added, neither GAA nor GAA-δ5 could be detected. Adding ornithine into the reaction significantly reduced the formation of both GAA and GAA-δ5.

When measuring ornithine in each of the different reaction combinations, we observed that its concentration was similar across all of the reactions. Because of where the stable isotopes are located on the GLY-δ3 and ARG-δ2 (Figure 1), the reaction will result in all of the stable isotopes being present on the GAA, while the ornithine will have none. As such, it is expected that the location of the label will not affect the amount of ornithine being produced (Figure 3B). Since ornithine is an inhibitor of AGAT enzymatic activity, we also aimed to determine if the amount of ornithine produced by AGAT could inhibit its function. In order to assess this, we performed the reaction within mouse kidney homogenates and added different concentrations of ornithine to assess how this affected AGAT activity (Figure 3C). The activity of AGAT activity was plotted against ornithine concentration, and a one-phase decay curve was fit to the data and calculated an IC_50_ of approximately 1 mM. Based on the data, we can see that the ornithine concentration produced within the reaction, approximately 40 µM, may have a negligible inhibitory effect on AGAT. As such, we believe that this assay allows for an accurate determination of AGAT enzyme activity.

### 2.2. Identifying the Optimal Reaction Conditions for the Enzymatic Assay

Following the validation of the assay, our next objective was to determine the optimal conditions for measuring AGAT enzymatic activity on the basis of temperature, pH, and duration.

In order to determine the optimal temperature and pH conditions for measuring AGAT activity, we performed the assay across a range of temperatures (4–75 °C) and pH conditions (6–10). The levels of GAA-δ5 were quantified and normalized to tissue weight in order to determine enzyme activity. These data were then plotted against temperature or pH and fitted to a Log-normal of Gaussian curve, respectively. Based on the data, the ideal temperature and pH were calculated as 37 °C and 7.6, respectively (Figure 4A,B).

To identify the optimal duration for the assay, we performed the reaction for 30, 60, 90, and 120 min and calculated the concentration of GAA-δ5 at each timepoint. By plotting the GAA-δ5 levels against reaction time, we observed a strong linear correlation between assay duration and GAA-δ5 production (Figure 4C). This linear trend suggests that the substrate concentration used in this assay is in excess and that there is minimal risk for substrate depletion when performing the assay for 30 to 120 min.

### 2.3. Determining Enzyme Kinetics of AGAT for Its Substrates

Our next objective was to determine the enzyme kinetics of AGAT by calculating K_m_ and V_max_ for each of its substrates, GLY-δ3 and ARG-δ2. In order to do so, we performed the assay in mouse kidney homogenate with a constant concentration of one substrate while varying the concentration of the other substrate. By plotting the enzyme activity against substrate concentration and fitting it to Michaelis-Menten kinetics, we determined the K_m_ and V_max_ for GLY-δ3 to be 2.06 mM (1.82 to 2.34, 95% CI) and 6.48 ± 0.26 pmol/min/mg kidney and that for ARG-δ2 to be 2.67 mM (1.99 to 3.60, 95% CI) and 2.17 ± 0.49 pmol/min/mg kidney, respectively (Figure 5).

### 2.4. Presence of Arginase Prevents Accurate Quantification of AGAT Activity in the Liver

When determining AGAT activity in several mouse organs, we initially observed very low perceived AGAT activity in liver at 0.003 pmol/min/mg tissue. Further investigation of the unexpectedly low perceived AGAT activity in mouse liver revealed an interesting observation with regards to interfering metabolites during the enzyme assay reaction.

When analyzing the levels of labeled substrates present following the reaction, we observed that the concentration of GLY-δ3 was relatively similar across all organs at a concentration of 10,000 µM (Figure 6A). However, the concentration of ARG-δ2 was significantly lower in the liver at around 2000 µM compared to the other organs which had approximately 5000 µM (Figure 6B). Furthermore, the level of ornithine in the liver was almost 100-fold higher than the other organs with a concentration at around 2500 µM (Figure 6C). The fact that the level of ARG-δ2 was significantly depleted while ornithine was elevated following enzyme assay raised the suspicion that the enzyme arginase 1 (ARG1) could interfere with AGAT activity measurement. ARG1 is highly expressed in the liver, but not in the other organs that we investigated. It acts by catalyzing the hydrolysis of L-arginine to form L-ornithine and urea. Ornithine inhibits AGAT enzyme activity. Thus, the ornithine formed by the action of arginase in the course of the enzyme assay explains why the perceived AGAT activity in the liver was much weaker than expected.

In order to test this, we performed the enzyme assay in mouse liver homogenates with different concentrations of Nor-NOHA which is a potent inhibitor of arginase. If the changes in the concentration of ARG-δ2 and ornithine were due to arginase activity, we would anticipate the levels of those metabolites to be similar to those in the other organ preparations upon the addition of Nor-NOHA. As expected, with an increase in the amount of Nor-NOHA in the enzyme reaction, the levels of ARG-δ2 increase until they plateau at around 5000 µM (Figure 6D), while the concentration of ornithine decreases to around 30 µM (Figure 6E), which is similar to the other organs. Addressing the high levels of arginase in the liver by inhibiting its activity with Nor-NOHA, we can more accurately determine the activity of AGAT in the liver to be approximately 0.4 pmol/min/mg tissue (Figure 6F).

### 2.5. Quantification of Specific AGAT Activity in Mouse Tissues

We investigated AGAT activity in wild-type mouse organs including liver, kidney, brain, and muscle. Our results showed that mouse kidneys had the highest levels of enzymatic activity followed by brain and liver with 4.6, 1.8, and 0.4 pmol/min/mg tissue, respectively (Figure 7). Both the heart and muscle had no detectable levels of AGAT activity. In addition to measuring enzyme activity in wild-type mouse tissues, we also used this assay in AGAT knockout mice. As expected, we were unable to detect AGAT activity in any of the organs.

### 2.6. Quantification of Specific AGAT Activity in Patient Cell Lines

In order to assess the feasibility of using this enzyme assay to quantify AGAT activity in human subjects suspected of AGAT deficiency, we performed the assay in various immortalized human cell lines and human-derived samples. For the immortalized cell lines, we chose to use HEK293 and HeLa cells due to their ubiquitous use in research as well as HAP1, RH30, and HepaRG cells due to their high expression of AGAT. For the human-derived samples, we used leukocytes as well as lymphoblasts and fibroblasts, as they are cultured from blood and skin biopsies, respectively, which are more easily obtainable for diagnostic purposes compared to other biological materials.

From the immortalized cell lines, we found the highest levels of AGAT activity in RH30 cells with 92.3 pmol/min/mg protein, followed by HepaRG, HAP1, and HeLa cells with activities of 18.1, 11.3, and 0.3 pmol/min/mg protein, respectively (Figure 8). In lymphocytes from 3 healthy controls, the AGAT activities were found to be 11.3, 6.6, and 14.7 pmol/min/mg protein, respectively. In leukocytes, AGAT activity was 11.0 pmol/min/mg protein. In our assay, AGAT activity was not detectable in the HEK293 cells, human fibroblasts, and lymphoblasts of a patient with AGAT deficiency.

## 3. Discussion

The synthesis of creatine within the body is an essential function of energy homeostasis, as it allows for the generation of phosphocreatine which facilitates the regeneration of ATP from ADP. Biosynthesis of creatine within the body is especially vital during infancy and early childhood as the body and brain are developing and require high levels of energy to fuel growth and differentiation. Failure to maintain an adequate level of creatine can occur due to deficiencies in either AGAT or GAMT, which are involved in the biosynthesis of creatine. In addition, a defective creatine transporter (CT1) can also result in low levels of intracellular creatine due to an inability to import it into cells. These deficiencies, collectively referred to as CDS, are characterized by developmental delays, intellectual disabilities, behavioral abnormalities, speech problems, and seizures.

The diagnosis of CDS relies on the measurement of creatine levels via MRS and tandem mass spectrometry or through genetic sequencing to identify mutations within the genes that encode for AGAT, GAMT, or CT1. We have optimized and validated the use of an enzyme assay that allows for the measurement of AGAT activity in various tissues and human cells.

In this study, our goal was to use a stable isotope-labeled substrate assay in order to measure AGAT activity from different materials such as tissues and cells. To do so, we investigated the specificity of the assay for the formation of GAA and GAA-δ5. We confirmed that both ARG-δ2 and GLY-δ3 were required for the formation of GAA-δ5 as replacing one of the labeled substrates with an unlabeled counterpart did not allow for its production. While unlabeled GAA could be used to determine AGAT enzyme activity, GAA-δ5 provides a more specific approach since it avoids any potential interference from any GAA that might be already be present in a sample.

We modified the assay of Verhoeven et al. [21] by decreasing the concentration of substrates and reducing the reaction volume to make the enzyme assay compatible with a 96-well plate. After further optimizing the reaction conditions by taking into account how the temperature, pH, and duration of the condition affected the activity of AGAT, we applied the enzyme assay in mouse tissues.

We determined the enzyme kinetics of AGAT in mouse kidneys; these organs were used due to their high endogenous expression of AGAT. While there have not been any previous studies that have quantified the K_m_ and V_max_ of AGAT in mice, the data that we have obtained are comparable to those obtained from human and rat kidneys [19].

While measuring the AGAT activity in various mouse tissues with the goal of establishing a comprehensive baseline of enzyme activity in each organ, we were able to observe that the levels of enzymatic activity correlated fairly well with the amount of AGAT protein. The only exception to this trend was the liver, which had persistently low perceived levels of AGAT activity. In the context of creatine synthesis, the kidney is known primarily for its high expression of AGAT while the liver has an abundance of GAMT [7]. Nonetheless, in many mammals such as humans, cows, and monkeys, the liver also contains significant levels of AGAT, on both RNA and protein levels [7]. As such, we were surprised that the liver had barely detectable levels of AGAT activity. However, further investigation revealed that the liver contains high levels of ARG1, which acts to catalyze the hydrolysis of L-arginine-generating L-ornithine and urea. Due to the presence of arginase, this affects AGAT activity in two ways: a depletion in the amount of labeled ARG-δ2 that can be used to generate GAA-δ5 and an increase in the concentration of ornithine, which acts to inhibit the function of AGAT. Because of these two effects, it is conceivable that initially the liver showed such low levels of AGAT activity.

AGAT activity is markedly reduced in the presence of high concentrations of ornithine (15 mM), resulting in an approximately 95% reduction in GAA and GAA-δ5 formation which corroborates previous studies [21]. The quantification of ornithine concentrations shows similar levels across all reaction conditions. The synthesis of ornithine within cells can occur due to the activity of either AGAT or ARG1. While AGAT synthesizes ornithine as a by-product of the formation of GAA via transfer of the amidino group of L-arginine to L-glycine, ARG1 cleaves L-arginine to generate L-ornithine and urea. When taking into account the use of ARG-δ2 in the reaction, both AGAT and ARG1 produce a non-labeled ornithine. As such, there should be no difference in the amount of ornithine produced whether unlabeled arginine or labeled arginine is present, which is what we observed. Furthermore, while AGAT produces ornithine in this assay, the amount of it that is generated is believed to have a minimal effect on AGAT activity.

In order to deal with the high levels of ARG1, an arginase inhibitor, Nor-NOHA, was added into the reactions that used liver homogenate. In the presence of Nor-NOHA, we observed that the concentrations of both ARG-δ2 and ornithine returned to levels that were comparable to the other organ homogenates leading to accurate and reliable quantifications of the enzymatic activity of AGAT in the liver.

In addition, when measuring the AGAT activity in various mouse tissues, we determined that kidney had the highest AGAT activity followed by brain and liver while the heart and muscle contained no detectable AGAT activity.

To test whether this enzymatic assay could be used for diagnostic purposes, we measured AGAT activity in human leukocytes, lymphoblasts, and fibroblasts. AGAT activity was readably detected in lymphoblast cell lines from healthy subjects and also, but to a lower degree, in a pooled leukocyte pellet. In addition to these control human cells, we were also able to obtain a lymphoblast cell line from a patient with AGAT deficiency. As expected, there was no AGAT activity in this sample. Finally, with the fibroblast cells, we were unable to detect AGAT activity in them. We attempted to optimize the enzyme assay for fibroblasts by increasing both the number of cells and substrate concentrations, but we were still unable to detect AGAT activity. In summation, our results indicate that this assay quantifies AGAT activity in leucocytes and lymphoblasts and allows for discrimination between samples derived from healthy controls and patients. Based upon our results, leukocytes would be feasible, but lymphoblasts would represent the ideal cells for quantifying AGAT activity.

In addition to using this assay for diagnostic purposes, we assessed the assay for the purposes of drug screening with AGAT as the druggable target [14]. As such, we measured AGAT activity in several immortalized human cell lines commonly used in research laboratories, such as HEK293, HeLa, RH30, HepaRG, and HAP1. Our data showed that RH30 cells had the highest levels of AGAT activity, five times higher than HepaRG, followed by HAP1 and HeLa cells; HEK293 had no detectable levels of AGAT activity. Therefore, AGAT activity can be reliably quantified in most cell lines, suggesting that they could be used for potential drug screening studies in the future.

Moving forward, we plan to adapt this assay to quantify the concentration of labeled creatine produced in order to measure GAMT activity. This will increase the applicability of this assay, as it would allow for simultaneous quantification of both enzymes that are involved in the biosynthesis of creatine.

In conclusion, we have validated and optimized an enzyme assay that reliably measures AGAT activity in a wide range of samples, and we resolved a confounding factor in the liver that affected the ability of the assay to produce accurate results.

## 4. Materials and Methods

### 4.1. Cells and Tissues

Lymphocyte cell lines from three healthy controls (Control 1-3/GM13072, GM14983, GM14926 respectively) and one patient with AGAT deficiency (AGAT D-/GM27955) were purchased from Coriell Institute for Medical Research (Camden, NJ, USA). Leukocytes were pooled from leftover pellets from pediatric samples prepared for clinical enzyme assays (The Hospital for Sick Children, Toronto, ON, Canada). Fibroblasts were from an in-house laboratory control cell line. HeLa and HEK293 were from American Type Cell Culture (ATCC; Manassas, VA, USA); HAP1 cells were from Horizon Discovery LTD (Cambridge, UK); RH30 were purchased from ATCC via Cedarlane (Burlington, ON, Canada); and HepaRG were purchased from ThermoFisher (Ottawa, ON, Canada). Mouse tissues were from in-house C57BL/6 wild-type mouse colonies.

### 4.2. Cell Culture Condition

Lymphocyte cells were cultured to confluency in T25 flasks with RPMI-1640 that was supplemented with 15% fetal bovine serum (FBS). Fibroblasts were grown to confluency in 10 cm plates using AMEM with 10% FBS. For immortalized cell lines, HEK293 and HeLa cells were cultured in DMEM with 10% FBS, HAP1 cells were grown in ISCOVES with 10% FBS, RH30 cells were grown in RPMI-1640 containing 10% FBS, and HepaRG cells were grown in William’s E media supplemented with 10% FBS, 0.14 units/mL insulin (Humulin-R), and 50 µM hydrocortisone.

### 4.3. Chemicals and Reagents

Chemicals for the experimental reactions and applications were: potassium phosphate mono- and dibasic, ammonium formate (Sigma-Aldrich Canada Co., Oakville, ON, Canada), formic acid LC/MS-grade and methanol HPLC-grade (Fisher Scientific, Ottawa, ON, Canada), trichloroacetic acid (VWR International, Radnor, PA, USA), and 3M butanol·HCl (Regis, Morton Grove, IL, USA). Reagents for calibrators and internal standards were: glycine (13C2, 99%; 15N, 99%), L-arginine:HCL (guanidino-15N2, 98%+), glycine (1-13C, 99%) (Cambridge Isotopes, Tewksbury, MA, USA), L-arginine (ARG), glycine (GLY), ornithine (ORN), guanidinoacetic acid (GAA), guanidinoacetic acid-d2 (GAA d2), ornithine-d6 (OR d6), arginine-d7 (ARG d7) (Sigma-Aldrich Canada Co., Oakville, ON, Canada). Tubes for mouse tissue homogenisation: VWR 2 mL × 2.8 mm Ceramic Hard tissue Homogenizing Mix and VWR 2 mL × 1.4 mm Ceramic Soft Tissue Homogenizing Mix were from VWR (VWR International, Radnor, PA, USA).

### 4.4. Liquid–Chromatography Tandem Mass Spectrometry (LC–MS/MS)

LC-MS/MS was carried out using an Exion LC AD UHPLC system coupled with QTRAP 6500plus (AB Sciex LLC, Framingham, MA, USA). The separation of metabolites was performed using gradient binary elution at a flow rate of 0.7 mL/min and a temperature at 45 °C on a Kinetex C18 100 Å, 5 µm, 100 × 4.6 mm LC column (Phenomenex Inc., Torrance, CA, USA). Solvent A consisted of 0.5 mmol/L ammonium formate, 0.1% (*v*/*v*) formic acid in water and solvent B consisted of 0.5 mmol/L ammonium formate, 0.1% (*v*/*v*) formic acid in methanol. The mobile phase was used at 100% A at 0 min; 100% B at 5.0 min; 100% B at 7.5 min; 100% A at 7.55 min; 100% A at 10 min. The sample injection volume was 1 µL. The detection was performed in the positive ionization and multiple reaction monitoring (MRM) scan mode using ion source parameters of TEM—600 °C, de-clustering potential—60.0, capillary voltage—5500 V, curtain gas—30, GS1—30, and GS2—20. The optimal ion transitions for the analytes and their retention times are shown in Table 1.

### 4.5. Data Analyses

The data were processed and analyzed by Analyst 1.7.0 software (AB Sciex LLC, Framingham, MA, USA). The calibration curves were generated from the ratio of analyte to the IS peak using linear fit and weighting of 1/x. GAA calibrators were used to calculate concentration of GAA-δ5.

### 4.6. Calibrators and Internal Standard (IS) for LC-MS/MS

Stock solution of calibrators and internal standards were prepared in Milli-Q water and aliquots were stored at -20 for a year. The concentrations of the working calibrators were 500, 250, 100, 50, 25, 10, 5, 2.5, 0 µM for ARG, ARG-δ2, ORN; 10, 5, 2, 1, 0.5, 0.25, 0.1, 0.05, 0 µM for GAA; 1000, 500, 200, 100, 50, 20, 10, 5, 0 µM for GLY and GLY-δ3. The internal standard contained a mixture of ORN d6, ARG d7 at 100 µM, GAA d2 at 10 µM, and GLY 13C_2_ at 200 µM.

### 4.7. Preparation of Mouse and Cell Samples for Enzymatic Assay

In total, 50–100 mg of frozen mouse kidney, heart, or muscle were cut on ice and transferred into a pre-chilled 2 mL tube with 2.8 mm ceramic beads; liver or brain tissues with an approximate mass of 100 mg were transferred into a 2 mL tube containing 1.4 mm ceramic beads. After addition of 0.5–1 mL of a cold 0.1 M potassium phosphate buffer, pH 7.4, samples were homogenized on Omni Bead Ruptor Elite using program of 5.65 m/s, 2 cycles of 1 min, 10 sec dt for hard tissues, and 4.85 m/s for 1 cycle of 20 s for soft tissues. When assessing the effect of pH on enzyme activity, tissues were homogenized in 1 mL of water. Tissue homogenate was used immediately or stored at −80 until needed. Frozen leukocyte pellets were resuspended in 500 µL of 0.1 M potassium phosphate buffer, pH 7.4. Cultured immortalized cell lines, lymphocytes, and fibroblast cells were washed with PBS, pelleted, and resuspended in 200–700 µL of 0.1 M potassium phosphate buffer, pH 7.4. All samples were then sonicated in 2 cycles for 10 s. Cell lysates were used immediately.

### 4.8. Preparation of Samples for LC-MS/MS

Following enzyme assay, protein precipitation in cell lysates or tissue homogenates was carried out by treating 100 µL of each sample with 25 µL of 30% TCA, followed by vortexing and centrifugation at 18,000× *g* for 10 min. Clear supernatants in amounts of 100 µL were mixed with 10 µL of IS and 500 µL of methanol. Samples were then vortexed and centrifuged for 5 min at 18,000× *g*. Supernatant was collected, transferred to a glass tube, and evaporated under nitrogen gas for 30 min or until dry. The derivatization step was carried out through addition of 100 µL of butanol HCl to a dry residue, followed by vortexing and incubation at 60 °C for 30 min. After cooling to room temperature, derivatized samples were evaporated under nitrogen gas for 30 min and finally resuspended in 700 µL of methanol.

### 4.9. AGAT Enzyme Assay Specificity Assessment

All substrates were diluted in water. Substrate combinations and stock concentrations were as follows: 37.5 mM GLY + 37.5 mM ARG, 37.5 mM GLY + 37.5 ARG-δ2, 37.5 mM GLY-δ3 + 37.5 mM ARG, 37.5 mM GLY-δ3 + 37.5 mM ARG-δ2, 37.5 mM GLY + 37.5 mM ARG + 75 mM ORN, 37.5 mM GLY-δ3 + 37.5 mM ARG-δ2 + 75 mM ORN. The enzyme assay consisted of 50 µL of mouse kidney homogenate, 50 µL of substrate mixture, 75 µL water, and 75 µL 0.1 M potassium phosphate buffer, pH 7.4. The reactions were incubated at 37 °C for 1 h. Following incubation, samples were prepared for LC-MS/MS.

### 4.10. Effect of Ornithine Dose Response on AGAT Activity

Mouse kidney tissues were prepared for the enzyme assay. The enzyme reaction consisted of 50 µL of mouse kidney homogenate; 50 µL of substrate mixture containing 37.5 mM GLY-δ3, 37.5 mM ARG-δ2, and 0.001 to 50 mM ornithine; 75 µL water; and 75 µL 0.1 M potassium phosphate buffer, pH 7.4. The reactions were incubated at 37 °C for 1 h. Following incubation, samples were prepared for LC-MS/MS.

### 4.11. Determination of K_m_ and V_max_ for AGAT

GLY-δ3 and ARG-δ2 substrates were serially diluted in water to the following concentrations: 150, 75, 37.5, 18.75, 9.38, 4.69, 2.34, 1.17, 0.56, 0.29, 0.15, 0.07, and 0.04 mM. To determine K_m_ and V_max_ of AGAT for GLY-δ3, substrate mixtures consisting of 9.38 mM ARG-δ2 and GLY-δ3 were created at the above-mentioned concentrations. To determine K_m_ and V_max_ of AGAT for ARG-δ2, substrate mixtures consisting of 9.38 mM GLY-δ3 and ARG-δ2 were created at the above-mentioned concentrations. The enzyme assay consisted of 50 µL of mouse kidney homogenate, 50 µL of substrate mixture prepared above, 75 µL of water, and 75 µL of 0.1 M potassium phosphate buffer, pH 7.4. The reaction was incubated at 37 °C for 1 h. Following incubation, samples were prepared for LC-MS/MS.

### 4.12. Temperature Dependence

Mouse kidney tissues were prepared for the AGAT enzyme assay. The reaction consisted of 50 µL of kidney homogenate, 50 µL of substrate mixture containing 9.4 mM GLY-δ3 and 9.4 mM ARG-δ2, 75 µL water, and 75 µL of 0.1 M potassium phosphate buffer, pH 7.4. The reaction was incubated at one of the following temperatures for 1 h: 4 °C, 22 °C, 37 °C, 42 °C, 55 °C, and 75 °C. Following incubation, samples were prepared for LC-MS/MS.

### 4.13. pH Dependence

Mouse kidney tissues were prepared for the AGAT enzyme assay. The reaction contained 50 µL of kidney homogenate, 50 µL of substrate mixture containing 9.4 mM GLY-δ3 and 9.4 mM ARG-δ2, 25 µL water, and 125 µL of 0.1 M potassium phosphate buffers at one of the following pH: 6.0, 6.5, 7.0, 7.4, 8.0, 9.0, 10.0. The reaction was incubated at 37 °C for 1 h. Following incubation, samples were prepared for LC-MS/MS.

### 4.14. Time Dependence

Mouse kidney tissues were prepared for the AGAT enzyme assay. Reaction contained 50 µL of kidney homogenate, 50 µL of substrate mixture containing 9.4 mM GLY-δ3and 9.4 mM ARG-δ2, 75 µL water, and 75 µL of 0.1 M potassium phosphate buffer, pH 7.4. The reaction was incubated at 37 °C for one of the following durations: 0 min, 30 min, 60 min, 90 min, or 120 min. Following incubation, samples were prepared for LC-MS/MS.

### 4.15. Quantification of AGAT Activity in Mouse Tissues

The reaction consisted of 50 µL of tissue homogenate, 50 µL of substrate mixture containing 37.5 mM GLY-δ3 and 37.5 mM ARG-δ2, 75 µL of water, and 75 µL of 0.1 M potassium phosphate buffer, pH 7.4. For mouse liver samples that were used to test efficacy of Nor-NOHA, the 75 µL of water was replaced with an equivalent volume of Nor-NOHA resuspended in water at concentrations ranging from 2 to 1500 µM. Final concentration of Nor-NOHA used for quantifying AGAT activity in the liver was 500 µM. The reaction was incubated at 37 °C for 1 h. Following incubation, samples were prepared for LC-MS/MS.

### 4.16. Quantification of AGAT Activity in Cell Samples

Cell samples were prepared for the AGAT enzyme assay. The reaction contained 50 µL of cell lysate, 50 µL of substrate mixture containing 37.5 mM GLY-δ3 and 37.5 mM ARG-δ2, 75 µL of 0.1 M potassium phosphate buffer, pH 7.4, and 75 µL of water. The reaction was incubated at 37 °C for 1 h. Following incubation, samples were prepared for LC-MS/MS. Immediately after performing the AGAT enzyme assay, 50 µL of the reaction was collected for quantification of protein concentration using the Pierce™ BCA Protein Assay Kit (ThermoFisher, Etobicoke, ON, Canada, #23225) according to the supplied protocol.

## Figures and Tables

**Figure 1 ijms-25-12490-f001:**
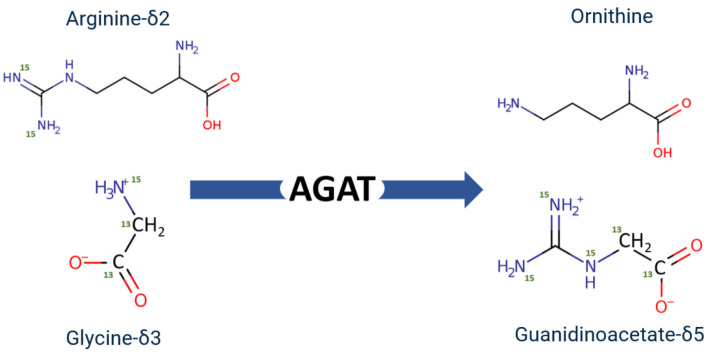
Schematic of the synthesis of ornithine and guanidinoacetate from GLY-δ3 and ARG-δ2. Green numbers indicate the locations of stable isotopes.

**Figure 2 ijms-25-12490-f002:**
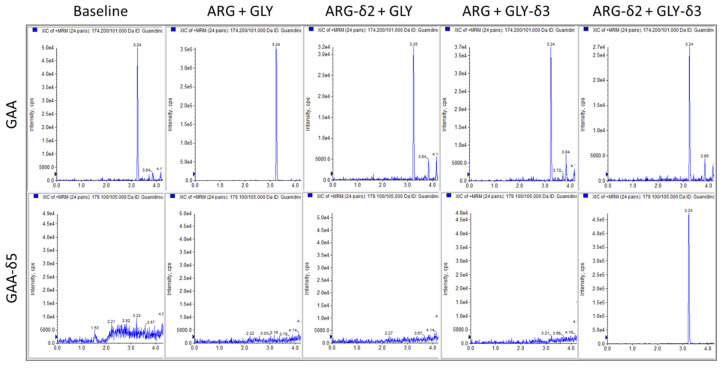
GLY-δ3 and ARG-δ2 allow for the formation of GAA-δ5. Combinations of either unlabeled (ARG + GLY), one labeled (ARG-δ2 + GLY and ARG + GLY-δ3), or both labeled substrates (ARG-δ2 + GLY-δ3) at a final concentration of 30 mM were used for the AGAT enzyme reaction in mouse kidney lysates. GAA is present in kidney tissue endogenously, and its formation is observed in all aforementioned reactions. However, the formation of GAA-δ5 only occurs when both labeled substrates are present. Top panel shows the appearance of GAA peak (+MRM, transition 174.2/101) and lower panel GAA-δ5 (+MRM, transition 179/105).

**Figure 3 ijms-25-12490-f003:**
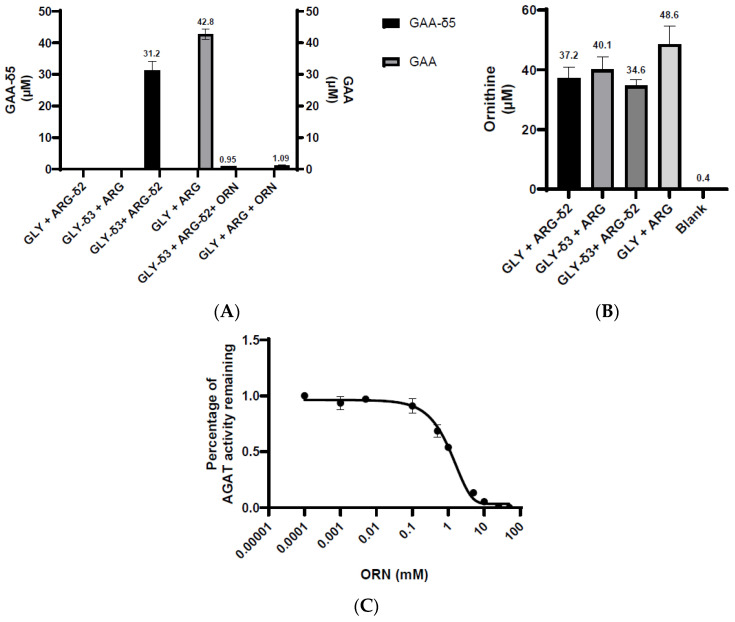
The formation of GAA-δ5 requires both labeled substrates. (**A**) Mouse kidneys were homogenized in potassium phosphate buffer, pH 7.4 and were incubated with different combinations of unlabeled and labeled substrates at a concentration of 7.5 mM in conjunction with 15 mM ornithine, and the concentrations of both GAA and GAA-δ5 were measured in positive-ionization MRM with the following transitions GAA 174.2/101, GAA-δ5 179/105, ornithine 189.2/70.1). The formation of GAA occurred only when both unlabeled substrates were present, while the synthesis of GAA-δ5 was exclusive to having both labeled substrates. In both reactions, the addition of ornithine resulted in a significant reduction of GAA and GAA-δ5 being produced. (**B**) Across all the different substrate combinations, the concentration of ornithine produced was relatively similar. (**C**) Dose–response curve showing the effect of ornithine on AGAT activity. Based on the concentration of ornithine produced in the enzyme reaction (**B**), it does not seem to have an inhibitory effect on AGAT activity. 0.0001 = 0 mM ornithine concentration.

**Figure 4 ijms-25-12490-f004:**
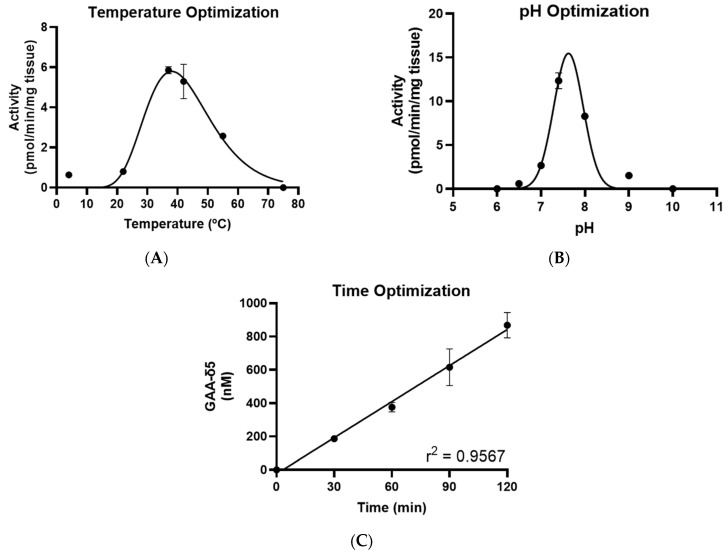
Optimization of reaction conditions for enzyme assay to measure AGAT activity. (**A**) Temperature. Mouse kidney samples were homogenized in potassium phosphate buffer, pH 7.4. Th reaction was performed using 1.8 mM of both labeled substrates, GLY-δ3 and ARG-δ2, for a duration of 1 h using the following temperatures: 4 °C, 22 °C, 37 °C, 42 °C, 55 °C, and 75 °C. GAA-δ5 was quantified to determine AGAT activity and was normalized to the amount of kidney used in the reaction. The data were fit to a Log-normal curve to visualize the correlation between temperature and AGAT activity. Observations showed peak activity to be approximately at 40 °C. to visualize the correlation between temperature and AGAT activity. Observations showed peak activity to be approximately at 40 °C. (**B**) pH. Mouse kidney samples were homogenized in water. Th reaction was performed using 1.8 mM of both labeled substrates, GLY-δ3 and ARG-δ2, for 1 h in potassium phosphate buffer under the following pH conditions: 6, 6.5, 7, 7.4, 8, 9, 10. AGAT activity was determined by normalizing GAA-δ5 concentrations to the amount of kidney used in the reaction. A Gaussian curve identified peak activity to occur when the pH of the environment was 7.6. (**C**) Time. Mouse kidney samples were homogenized in potassium phosphate buffer of pH 7.4. The reaction was performed using 1.8 mM of both labeled substrates, GLY-δ3and ARG-δ2, and incubated at 37 °C for durations of 0 min, 30 min, 60 min, 90 min, or 120 min. The concentration of GAA-δ5 was quantified and plotted against reaction duration. Results show a linear relationship between the duration of the enzyme assay and the amount of GAA-δ5 produced.

**Figure 5 ijms-25-12490-f005:**
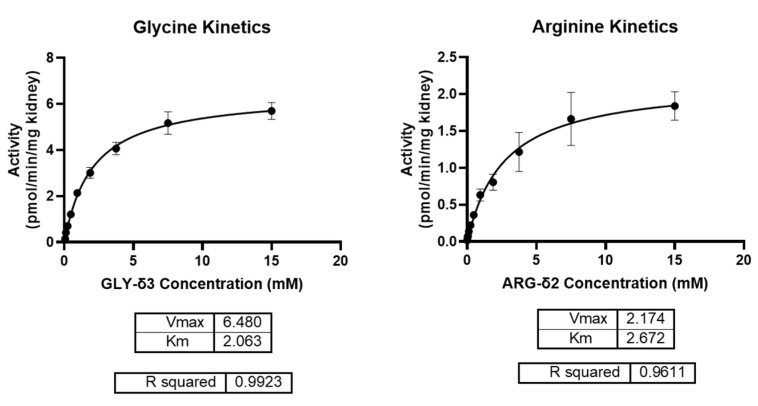
Determining enzyme kinetics of AGAT in mouse kidney. Mouse kidneys were homogenized in potassium phosphate buffer, pH 7.4. and homogenates were incubated with various concentrations of one labeled substrate while the other substrate was held at a constant concentration of 1.9 mM. The reaction was performed at 37 °C for 1 h. The concentration of GAA-δ5 was determined and normalized to the amount of kidney used in the reaction. Graphing the data and fitting them to a Michaelis–Menten curve shows that the K_m_ of AGAT for GLY-δ3 is 2.06 mM with a V_max_ of 6.48 ± 0.26. In comparison, the K_m_ of AGAT for ARG-δ2 is 2.67 mM with a V_max_ of 2.17 ± 0.49.

**Figure 6 ijms-25-12490-f006:**
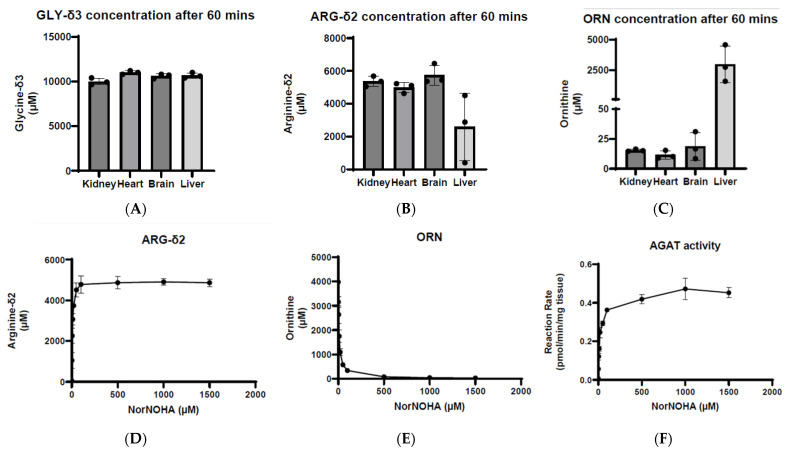
High levels of arginase in the liver can be inhibited through the use of nor-NOHA. Mouse tissues were homogenized in potassium phosphate buffer of pH 7.4 and were incubated with 7.5 mM of both GLY-δ3 and ARG-δ2 for 1 h at 37 °C. The concentrations of (**A**) GLY-δ3, (**B**) ARG-δ2, and (**C**) ornithine were compared across the various tissues. In the liver, the levels of ARG-δ2 were significantly lower, while the levels of ornithine were much higher when compared to the organs. In contrast, the concentration of GLY-δ3 was relatively constant across all tissues. In order to determine if nor-NOHA was an effective inhibitor of arginase activity, mouse livers were homogenized in potassium phosphate buffer, pH 7.4 and were incubated with various concentrations of nor-NOHA in combination with 7.5 mM of both GLY-δ3 and ARG-δ2 for 1 h at 37 °C. As the concentration of nor-NOHA within the reaction increased, the levels of (**D**) ARG-δ2 and (**E**) ornithine increased and decreased, respectively, to levels that were consistent with the other organs. (**F**) Quantification of AGAT activity shows that higher concentrations of nor-NOHA within the reaction increase AGAT activity until it plateaus at around 0.4 pmol/min/mg tissue.

**Figure 7 ijms-25-12490-f007:**
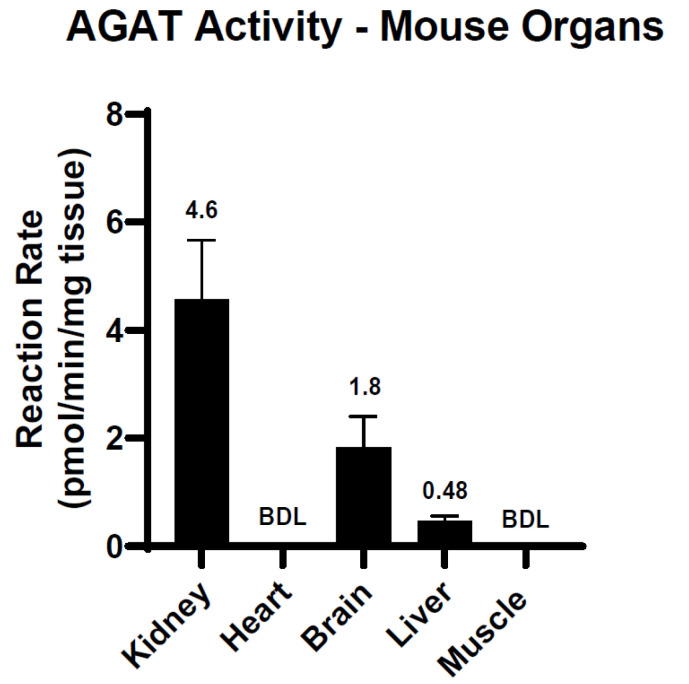
Determining AGAT activity in mouse organs. Mouse tissues were homogenized in potassium phosphate buffer, pH 7.4 and were incubated with 7.5 mM of both GLY-δ3 and ARG-δ2 for 1 h at 37 °C. The highest levels of AGAT activity were found in the kidney at 4.6 pmol/min/mg tissue, followed by the brain and liver. AGAT activity in the heart and muscle were below the detection limit (BDL). Reactions in liver samples were supplemented with 500 µM of nor-NOHA.

**Figure 8 ijms-25-12490-f008:**
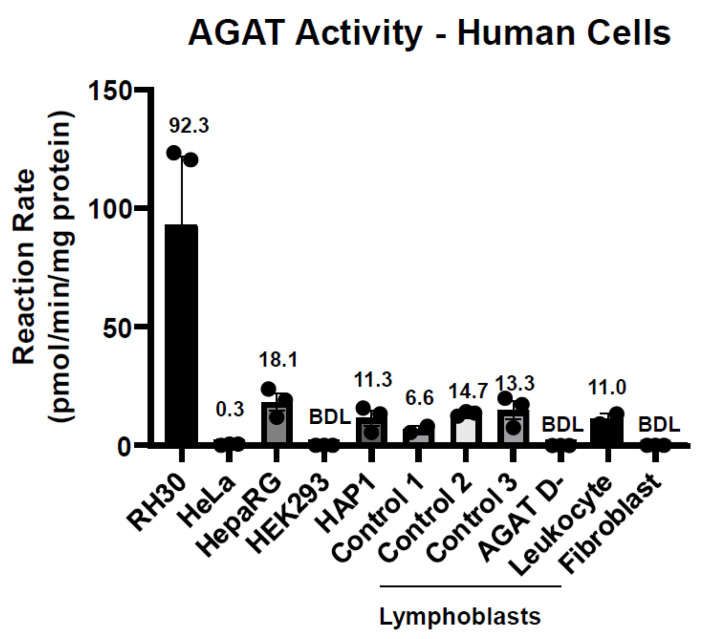
Measuring AGAT activity in cell lines and patient derived samples. Cells were sonicated in potassium phosphate buffer, pH 7.4, and lysate was incubated with 7.5 mM of both GLY-δ3 and ARG-δ2 for 1 h at 37 °C. Penta-GAA levels were normalized to protein concentrations in order to determine AGAT activity in each sample. From the immortalized cell lines, RH30 cells had the highest levels of activity followed by HepaRG, HAP1, and HeLa cells. In the patient derived cells, AGAT activity was detected in all lymphocyte cells with the exception of AGAT D-, as it derived from a patient that is deficient in AGAT. The leukocyte sample had AGAT activity that was slightly lower than the lymphocyte cells. In addition, fibroblasts were also observed to have no detectable amounts of AGAT.

**Table 1 ijms-25-12490-t001:** LC-MS/MS analytical conditions.

Analyte	Retention Time	Ion Transition (*m*/*z*)	CE	CXP
ARG	2.40	231.1/116.3	22	18
GLY	2.93	132.1/76.0	13	10
ORN	2.15	189.2/70.1	29	11
GAA	3.27	174.2/101	22	15
ARG 15N_2_	2.40	233.2/172	21	16
GLY 13C_2_15N	2.93	135.1/78.6	12	10
GAA 13C_2_15N_3_	3.27	179.1/105.0	18	20
ARG d7 (IS)	2.40	238.2/77.1	21	16
GLY 13C (IS)	2.93	133.1/77.0	13	10
ORN d6 (IS)	2.15	195.2/76.1	26	20
GAA d2	3.27	176.2/103.1	19	12

## Data Availability

The original contributions presented in this study are included in the article. Further inquiries can be directed to the corresponding author.

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
