# Peer review of "Validation and Optimization of a Stable Isotope-Labeled Substrate Assay for Measuring AGAT Activity"

_ijms, 2024, doi:10.3390/ijms252312490_

Round 1
Reviewer 1 Report
Comments and Suggestions for Authors
The experiments described in this manuscript demonstrate the systematic efforts by the authors to establish a method to determine the enzymatic activity of arginine: glycine amidinotransferase (AGAT), a crucial enzyme in creatine biosynthesis. Through these efforts, the authors addressed at least two confounding factors in the assay, including the contamination of endogenous metabolites by using stable isotope-labeled substrate (glycine and arginine), and the interference from hepatic arginase by using its inhibitor (NorNOHA). Nevertheless, the observed enzymatic activity of AGAT (unit in pmole/min/mg) is rather low compared to other common metabolic enzymes, such as the ones in urea cycle (as their unit, commonly in nmole/min/g). It is known that ornithine, the product of AGAT, is an allosteric inhibitor of AGAT. Therefore, this potential confounding factor on whether the AGAT activity in the described assay was suppressed by ornithine (especially at high-concentration incubations) was not explored in experiments or discussion.
Other issues and comments:
Line 56: Please elaborate on why AGAT is a therapeutic target for GAMT deficiency.
Line 92 & 93: Please provide a reference on the statement of “GAA is produced exclusively by AGAT”.
Line 115-120: The legend of Figure 2 can be improved for better clarity to help readers to understand the nature of enlisted values of GAA and labeled GAA in individual panels .
Line 196: change “Michaelis Menten” to “Michaelis-Menten”
Line 199: The authors mentioned they used fixed concentration of one substrate and used various concentrations of the second substrate as follows; 150, 75, 37.5, 18.75, 9.38, 4.69, 2.34, 1.17, 0.56, 0.29, 0.15, 0.07, and 0.04 mM (lines 483&484). The points in hyperbola graphs do not match with these numbers. Please double check.
Line 325&326: Please provide a reference for the statement “In the context of creatine synthesis, the kidney is known primarily for its high expression of AGAT while the liver has an abundance of GAMT”.
Table 1: What do the values of MRM ion transition pairs correspond to since they don’t reflect the expected m/z of positively charged ions?
Reviewer 2 Report
Comments and Suggestions for Authors
Summary: This manuscript describes a method for assaying the activity of L-arginine:glycine amidinotransferase (AGAT) using stable-isotope labelled precursors of glycine and arginine. This approach is an adaptation of a previously published method by Verhoeven et al. that used labelled glycine and guanido arginine precursors. In tissues with arginase activity such as the liver, an arginase inhibitor needs to be used otherwise the AGAT activity is under-reported.
Major points:
1) The authors need to better justify why their protocol is better than that of Verhoeven et al. because they seem to be identical.
2) Why can´t the activity be measured using the labelled glycine alone and measuring the appearance of m+3 GAA? This assay would be less expensive in terms of stable isotope precursors and would be insensitive to the arginase reaction.
Reviewer 3 Report
Comments and Suggestions for Authors
First, a general comment about Vmax. The maximal rate for an enzyme is the rate measured at saturation concentrations, in this case or glycine and arginine. Vmax is expressed in units of concentration per unit time (nmol/min/mL or similar) and it should be proportional to the concentration of the enzyme. Because this study uses tissue extracts, the concentration of the AGAT enzyme is not known (though it could be determined via western blots). Thus, the ratios of enzymatic rates to mg of tissues (pmoles/min/mg) are not rates but maximal specific activities, again if measured at saturation conditions. This is doubly confusing because in the graphs the specific activities are called reaction rates, which they are not, and in the text they are alternatively referred to as Vmax or just "activity". The former is not correct, but the latter is.
The authors mentioned in line 84 that ornithine is not a suitable readout since there are other pathways that can form or consume the amino acid, but if the assay is designed with U-labeled arginine, the products would be U-labeled ornithine and Gly-d3. That would be a cheaper and more informative assay that can be easily monitored via LC-MRM-MS.
Figure 3 needs to show the products of the cross isotopic transfers (Arg+ Gly-d3 and Gly+Arg-d2) since they can be distinguished by their corresponding m/z values for Q1 and Q3. Also, the method needs to be indicated in the caption. In Figure 3b, the concentration of ornithine for the unlabeled Arg+Gly reaction appears to be significantly higher. Is this because it is the product of the reaction, or were the different columns not normalized for tissue amounts?
How does the Ki (50%) in Figure 3c compares with reported product inhibition constants of ornithine for AGAT. Is that parameter known?
The plot in Figure 4c looks very linear, why is the correlation coefficient so poor (~0.95)?
In lines 192-198, what is the uncertainty in the Km values for Gly and Arg?
The LC-MS/MS are referred throughout the paper, but not described until the materials section. Since the authors used an LC-MRM approach, they should have described in Figure 2 what the chromatograms are showing. I was under the impression that Figure 2 are extracted chromatograms for an LC-MS (one quadrupole) measurements, but I am not totally clear on this. The concentrations of Arg and Gly or their isotopologues are not indicated in the caption either. Since they used LC-MRM methodology, they could have added those to the transition list, the reaction between unlabeled glycine and Arg-d2 as well as gly-d3 with unlabeled Arg. One can assume the tissues and samples have low concentrations of free-amino acids, but how do we know that the background reactions are not competing with the isotopically enriched AGAT activity?
Ornithine is one of the products of the reaction, it is expected to be a competitive inhibitor with respect to Arg. The guanidino transfer reaction is likely a reversible reaction, so the inhibition constant is probably reflecting a backflow transfer from labeled glycine to ornithine. This means that the measurements might be underestimating the true forward catalysis activity of AGAT. It would have been very useful to calculate what that back rate might be for the mouse tissues. The ornithine concentration can also be determined for these samples, using the same MRM method that was applied here. Line 301 needs an explanation of what "MRS" is.
In line 323, it is stated that the activity correlates well with the levels of the AGAT protein. That is the most common case, but the authors are not providing a measure of the AGAT levels (Western Blot or targeted proteomics). That would actually augment the value of this study as it should show not only the specific activity of the tissues but also the specific activity of AGAT in those tissues.
Round 2
Reviewer 1 Report
Comments and Suggestions for Authors
The revision has addressed all my comments.
Reviewer 2 Report
Comments and Suggestions for Authors
The revised manuscript addresses my critiques, thank you